# Nanomachinery Organizing Release at Neuronal and Ribbon Synapses

**DOI:** 10.3390/ijms20092147

**Published:** 2019-04-30

**Authors:** Rituparna Chakrabarti, Carolin Wichmann

**Affiliations:** 1Molecular Architecture of Synapses Group, Institute for Auditory Neuroscience and InnerEarLab, University Medical Center Göttingen, 37075 Göttingen, Germany; 2Center for Biostructural Imaging of Neurodegeneration, University Medical Center Göttingen, 37075 Göttingen, Germany; 3Collaborative Research Center 889 “Cellular Mechanisms of Sensory Processing”, 37099 Göttingen, Germany; 4Collaborative Research Center 1286 “Quantitative Synaptology”, 37099 Göttingen, Germany; 5Auditory Neuroscience Group, Max Planck Institute for Experimental Medicine, 37075 Göttingen, Germany

**Keywords:** exocytosis, membrane fusion, synaptic vesicles, synaptic vesicle tethering, synaptic vesicle pools, active zone, release machinery, ribbon synapses, electron microscopy, electron tomography

## Abstract

A critical aim in neuroscience is to obtain a comprehensive view of how regulated neurotransmission is achieved. Our current understanding of synapses relies mainly on data from electrophysiological recordings, imaging, and molecular biology. Based on these methodologies, proteins involved in a synaptic vesicle (SV) formation, mobility, and fusion at the active zone (AZ) membrane have been identified. In the last decade, electron tomography (ET) combined with a rapid freezing immobilization of neuronal samples opened a window for understanding the structural machinery with the highest spatial resolution in situ. ET provides significant insights into the molecular architecture of the AZ and the organelles within the presynaptic nerve terminal. The specialized sensory ribbon synapses exhibit a distinct architecture from neuronal synapses due to the presence of the electron-dense synaptic ribbon. However, both synapse types share the filamentous structures, also commonly termed as tethers that are proposed to contribute to different steps of SV recruitment and exocytosis. In this review, we discuss the emerging views on the role of filamentous structures in SV exocytosis gained from ultrastructural studies of excitatory, mainly central neuronal compared to ribbon-type synapses with a focus on inner hair cell (IHC) ribbon synapses. Moreover, we will speculate on the molecular entities that may be involved in filament formation and hence play a crucial role in the SV cycle.

## 1. Introduction

Vesicular exocytosis, involving either synaptic vesicles (SVs; [1,2,3]) or secretory exosomes [4,5,6,7] is essential for neuronal function. Neurotransmission is the dominant mechanism of communication between neurons and effector cells. It relies on Ca^2+^-dependent exocytosis of SVs, which takes place at dedicated AZs within the presynaptic terminals of neurons [8,9,10,11], as well as in sensory cells such as hair cells and photoreceptors [12,13,14,15,16]. In general, AZs are characterized by a dense protein network, which mediates the recruitment and release of SVs to the release site [17,18,19,20,21,22]. The clustering of SVs in proximity to voltage-gated Ca^2+^ channels is one prerequisite for regulated neurotransmission and is a common feature of chemical synapses [16,23,24,25,26,27,28,29,30,31,32]. Neuronal synapses such as from the hippocampus exhibit structures termed as electron-dense projections [17,22,33]. These are described as pyramidal structures of ~50 nm [17], forming a regular network or grid at the AZ [34,35,36,37] (Figure 1A,A’). Further, at frog neuromuscular junctions (NMJs), an elaborate and regular organization of electron-densities are found [19,38,39]. Similar to vertebrate synapses, invertebrate synapses also often show electron-dense projections that are suggested to be functional analogues [17]; such as the fruit fly *Drosophila* T-bars found at NMJs and in the central nervous system [40,41,42,43,44] and knob-like appearing structures with emerging filaments at *C. elegans* NMJs [45,46,47,48]. However, these invertebrate structures differ in regards to their shape and size [17,49,50], as well as their molecular composition [22,24,50,51,52,53,54,55,56], and will not be the focus of this review.

While many important molecules are conserved across chemical synapses, the sensory ribbon-type synapses appear more specialized, where graded membrane potentials trigger exocytosis at these synapses [57,58,59,60,61,62,63]. Ribbon synapses are present in vertebrate sensory systems such as in auditory hair cells [22,64,65] (Figure 1B,B’), in vestibular hair cells [66,67,68], photoreceptors [69,70] and retinal bipolar cells [70,71]. Further, they are also found in lower vertebrate pinealocytes in the pineal gland [72], fish lateral lines [73,74], and electroreceptors [75,76], as well as in frog saccular or turtle [77,78] hair cells. They all share a structural specialization appearing as a large electron-dense projection, the synaptic ribbon, which can reach in the photoreceptor a size of several hundreds of nanometers, and this way is capable to cluster a large number of SVs [71,79,80]. Some ribbon-type synapses maintain the highest rates of exocytosis documented so far [81,82,83], releasing even up to hundreds of SVs per second at an individual synapse for an extended period of time [12,81,84].

Moreover, ribbon synapses exhibit an elaborate structure between ribbon and membrane, termed as ‘presynaptic density’ in hair cells [16] and ‘arciform density’ at photoreceptor ribbon synapses [85]. Both are serving as an anchorage for the ribbon and contain the AZ scaffolding protein bassoon [16,85,86,87,88,89]. Consequently, upon bassoon disruption, synaptic ribbons mostly lose their attachment to the presynaptic AZ membrane, as have been shown in the photoreceptor and cochlear IHCs [16,85,86].

Other presynaptic proteins substantially differ at ribbon synapses in general and especially at IHC ribbon synapses. Next to the ribbon specific protein RIBEYE [69], the neuronal priming factors from the Munc and calcium-activator protein for secretion (CAPS) family [90] along with neuronal soluble N-ethylmaleimide-sensitive factor attachment protein receptor (SNARE) proteins; synaptobrevins/vesicle-associated membrane proteins (VAMPs) 1–3, syntaxins 1–3 and soluble N-ethylmaleimide-sensitive factor (NSF) attachment protein (SNAP) 25 [91], are serving no apparent function in exocytosis in IHCs. Complexins exist at retinal ribbons, but different isoforms are present [92,93,94,95], while they are absent from IHCs [96]. Hair cell ribbon synapse function depends heavily on the multi-C_2_ domain protein otoferlin [81,90,97,98,99,100,101], which is not playing a role at neuronal synapses. Another striking line of difference is that neurons mainly employ P/Q- and N-type Ca_V_2.1 and 2.2 Ca^2+^ channels [102,103,104], while IHC AZs employ L-type Ca_V_1.3 and retinal ribbon synapses Ca_V_1.4 channels [89,105,106,107,108,109,110,111,112,113].

From the early days of synaptic research, electron microscopy (EM) has been instrumental in providing the required spatial resolution to investigate the nanostructure at neuronal as well as ribbon-type synapses. Application of 3D structural methods like serial-section 3D reconstruction, electron tomography (ET), conical and scanning transmission EM (STEM) tomography after conventional aldehyde fixation (CAF), have made it possible to analyze the vesicle pool dynamics in a broad range of neuronal but also ribbon synapses. For example ribbon synapses of frog saccular hair cells [114,115,116,117], murine IHCs [16,81,89,90,98,118,119,120,121,122,123,124], retinal bipolar cells [125], and photoreceptors [126,127,128] have been studied extensively.

Moreover, applying different methods for EM sample preparation such as freeze-etching, CAF or rapid freezing, exposed that SVs are associated with the AZ via delicate filamentous structures that appear at vertebrate NMJ, central neuronal and ribbon-type synapses [16,19,38,90,116,118,120,121,122,125,129,130,131,132,133,134,135,136,137,138,139] (Figure 1C,D). Notably, we will use ‘filaments’ as a generic term in this review for these structures. Two categories of presynaptic filaments are described across synapses and species: (i) Filaments that link SVs within the presynaptic terminal lumen to form a network, coined as interconnectors from now on (Figure 2A’,A’’) and (ii) filaments, commonly termed as tethers, which connect SVs to the AZ-membrane (Figure 2C–E) [16,90,116,118,120,121,122,125,129,130,131,132,133,134,135,136]. Notably, we will use ‘filaments’ as a generic term. Ribbon-type synapses show a third main category i.e., filaments connecting SVs to the ribbon (Figure 2B’,B’’) [64,115,116,118,120,122,125,128,130,136], which we will refer to as ‘ribbon-attached filaments’ in this review. Given such consistent appearance in the presynaptic lumen, the phenomena of SV tethering appears to be essential for exocytosis at neuronal [132,133,135,138,139] and ribbon synapses [16,90,116,121,122,125,136].

Some of the studies mentioned above on neuronal and ribbon synapses have relied on CAF. The usage of CAF limits the temporal precision required to confidently detect distinct stages of exocytosis due to the slow process of the chemical fixation [140,141]. Therefore, in recent years, high-pressure freezing (HPF) [142,143,144,145], followed by freeze-substitution (FS) [146,147,148,149] has become the methodology of choice, as synapses can be captured in their near-to-native state with larger tissue depth [90,120,121,122,128,132,135,136,139,150,151,152,153,154,155]. Moreover, the challenging method of cryo-ET has already been successfully applied to neuronal culture and synaptosome preparations, thereby enabling a view on hydrated samples [133,134,156,157,158].

In this review, we will do a comparative discussion on SV tethering at neuronal and ribbon synapses and how state-of-the-art EM methodologies have been instrumental in advancing our understanding of the AZ architecture. Further, we will present the most recent view on potential candidates for filament formation at neuronal and ribbon synapses.

## 2. Synaptic Vesicle Pool Organization at Synapses

The quanta of neurotransmission at synapses are SVs [9,159], which are recruited to the AZ membrane to fuse and release their content upon a stimulus. Classically, three functional SV pools are defined at neuronal synapses that govern neurotransmission; the readily releasable pool (RRP), the recycling pool and the reserve pool [160,161,162,163,164,165].

Electrophysiological analyses at neuronal synapses define the RRP as those SVs, which are released instantly upon Ca^2+^ influx [166,167,168]. Therefore, readily releasable SVs are proposed to be allocated in a fusion-ready docked state at release slots and are therefore defined to be in direct contact with the AZ membrane [39,132,150,151,169]. Upon depletion of the RRP, SVs are replenished from the recycling pool (10%–20% of all the SVs). Continuous intense stimulation that triggers the depletion of recycling vesicles finally recruits SVs from the large reserve pool (80%–90% of all the SVs) [160,161,162,163,170,171]. However, significant spatial intermixing makes it difficult to draw a clear boundary between the recycling and the reserve pool SVs [160,162].

At hair cell ribbon-type synapses, two components of exocytosis are recorded using capacitance measurements. Firstly, the fast exponential capacitance rise that corresponds to depletion of the RRP of SVs, followed by a second slower sustained capacitance rise [12,172]. At bipolar cells, the second phase terminates after the fusion of all the ribbon-equivalent vesicles, leading to the postulation that release sites are replenished from the SV pool at the ribbon [71,173,174]. However, at hair cells, the sustained capacitance raise persists for a longer time. To keep up with the continuous demand, the synaptic ribbon needs to be resupplied with new SVs several times within a short time interval [12,26,81,115,116].

Complementary ultrastructural studies at ribbon-type synapses have proposed two discernable morphological SV pools [98,99,115,116,120,122]. (i) The membrane-proximal pool of SVs and (ii) the ribbon-associated pool of SVs. The membrane-proximal-SVs are proposed to represent the RRP [86,173,175,176,177] (also known as ultra-fast vesicle release pool; [114,116]). The proximity of these SVs to the AZ membrane suggests that these SVs deplete during the rapid initial phase of exocytosis with the onset of depolarization [12,172,178]. Whereas ribbon-associated SVs, which are away from the AZ membrane, are thought to form the rapidly-releasable SV pool [116] that replenishes the membrane-proximal SVs [81,114,115,120], and this way is representing a reserve pool [99]. To replenish the RRP, transport of SVs along the ribbon surface might be required as suggested in several studies [114,122,125,175,179,180,181]. Additionally, SVs within 350 nm around the ribbon are proposed to belong to the sphere of influence; a location of SV reformation at IHC ribbon synapses [123,182]. However, the interpretation of the morphological correlates for the functional SV pools at ribbon-type synapses is still a matter of intense debate and might not be uniform for all types of ribbon synapses.

Finally, recent studies suggest that before docking, priming, and fusion at the AZ membrane, SVs undergo several preparatory steps that involve tether formation at neuronal synapses [133,134,135,183] and ribbon-type synapses [122]. The typical scenario emerges that at the AZ membrane first single, presumably longer tethers recruit SVs, while multiple short tethers bring them closer to the AZ to prepare SVs for docking, which finally leads to the release. In the following sections, we will point out the role of filaments starting with the SV organization within the cytoplasm. Thereafter we will discuss SV recruitment to the membrane and finally docking and priming together with potential candidate proteins for filament formation.

## 3. Filament Organization in the Cytoplasm at Neuronal Compared to Ribbon Synapses

EM techniques like freeze-etching have shown early on that the cytoplasm of neuronal presynaptic terminals is filled with electron-dense filaments, interconnecting SVs (Figure 1; Figure 2A–A’’) [129,131,184]. They are also appointed as ‘connectors’ [133,134] or ‘bridge-filaments’ [135] at neuronal synapses. In this review, we will use the term ‘interconnector’, since this term has been used for ribbon synapses as well, where they are reported across various species and preparations (Figure 2B–B’’) [120,122,130,136]. To date, two primary functions of these interconnectors have been proposed at neuronal and ribbon synapses alike: (i) Clustering of SVs to spatially organize SVs within the presynaptic terminal [122,129,131,132,134,135,137], and (ii) mobilizing SVs to the AZ membrane [122,129,132,134].

## 4. Interconnectors in Neuronal Synapses form a Network of SVs

Interconnectors vary in length and in terms of numbers per SV [132,134,135,138,185]. Hence, it is tempting to speculate that the presence of these structures morphologically defines SV classification into different vesicle pools or also release competencies within the presynaptic terminal [132,134].

Interconnectors have been broadly described for the first time at mammalian Purkinje cell dendritic spines and cerebral cortex synapses, frog NMJs and in the electric organ synapses of electric ray after quick freeze-etching [129,131]. Although the tissue depth limited these studies, excellent tissue preservation up to a depth of 20–30 nm could be achieved, thus allowing reliable qualitative analyses. Later on, a systematic quantification of interconnector lengths and numbers has been performed at the lamprey reticulospinal nerve terminals combining CAF and ET. Here, an average length of approximately 15 nm is reported with each SV exhibiting up to 12 interconnectors [186]. However, using the same methodology, fewer but longer interconnectors per SVs are observed at mammalian neuronal synapses. For instance, on average 2.8 interconnectors per SV with a mean length of 32 nm and 3–6 interconnectors per SVs with a mean length of 11 nm are present in cat calyx of Held [187] and rat axo-spinous synapses in neocortex and hippocampus [185], respectively.

In recent years, the usage of HPF and subsequent FS at vertebrate synapses [132,135,137,150,151,152,153,154] has allowed the observation of interconnectors in greater tissue depth with near-to-native structural preservation (Figure 1C). Consistent with the CAF/ET studies, variable interconnector length could be found. For instance, at dissociated rat hippocampal neurons a range of 11–78 nm are reported with an average of 30 nm [135], reasonably comparable to developing zebrafish NMJs (~27 nm) [137]. Like in Gustafsson et al. [186] one SV could be held by up to 12 interconnectors in the rat hippocampus [135]. In contrast, in mouse hippocampal slice cultures, only 1.5 interconnectors per SV have been found [132].

Furthermore, a detailed description of synaptosomes from the mouse cerebral cortex using cryo-ET provided an insight into hydrated and completely unstained samples. In these studies, interconnectors measure ~10 nm, but could as well span up to 40 nm. Here, approximately three interconnectors per SV could be visualized [133,134,157]. This way, clusters of 10–50 SVs are formed that exhibit an increased interconnector density with advancing distance away from the AZ membrane [134]. Conclusively, interconnectors mediate the formation of an elaborate network of structurally linked SVs within the presynapse. Since the clustering of SVs appears more complex under rest compared to synapses during activity [129,134,135,137], interconnectors might play a vital role in the SV pool organization.

## 5. Interconnectors and Ribbon-Attached Filaments at Ribbon Synapses Mediate SV Recruitment and Transport towards the AZ Membrane

At ribbon synapses so far the main focus regarding filament organization has been on filaments emanating from the synaptic ribbon to cluster a large amount of SVs to the ribbon itself (Figure 2B,B’) [115,116,117,119,120,122,128,130,136]. These filaments represent a hallmark of these synapses and will be termed as ‘ribbon-attached filaments’ from now on, but are also described as ‘links’ in the literature [136].

Quantifications revealed that at some ribbon-type synapses SVs are clustered within 30 nm from the synaptic ribbon such as in frog saccular hair cells or at retinal ribbon synapses [116,125]. In line with these observations, freeze-etched ribbon synapses of frog photoreceptor cells exhibit ribbon-attached filaments to be ~30 nm long [130]. Ribbon-attached filaments could also be visualized reliably by using CAF with subsequent STEM tomography as done for mouse retinal bipolar ribbon synapses [125]. Moreover, in frog saccular hair cells, CAF/ET revealed ribbon-attached filaments of ~20 nm length [115,116]. Possibly depending on the way of preparation, at IHC ribbon synapses ribbon-attached filaments appear to be slightly longer on average, at murine IHC ribbon synapses they measure ~32 nm at rest [122] and guinea pig IHC ribbon synapses ~27 nm [136]. Both studies used HPF/FS followed by ET.

Based on the physiological and ultrastructural data it is postulated that docked SVs at the AZ membrane are replenished from the pool of SVs that is associated with the synaptic ribbon, possibly by transporting the vesicles to the AZ membrane [82,114,115,125,175,179], quite similar to what has been shown in frog NMJs [38]. Moreover, live epifluorescence microscopy and total internal reflection fluorescence (TIRF) microscopy indeed revealed that SVs move towards the AZ membrane at the synaptic ribbons of goldfish and zebrafish retinal bipolar cells [175,179]. Two models may explain the mobility of SVs at ribbon synapses: (i) The conveyor belt model suggests that SVs are transported in an adenosine triphosphate (ATP)-dependent manner in association with motor proteins [173,180,181]. Whereas, (ii) the ‘crowd surfing’ model suggests that SVs move via diffusion, once bound within a 30 nm zone around the synaptic ribbon; wherein transient binding and unbinding of multiple-ribbon-attached filaments allows a movement of SVs towards the AZ membrane [122,125]. However, in the literature, both models are still under debate, and different ribbon-type synapses might not necessarily reveal a unifying mechanism of SV transport.

Nevertheless, the involvement of ribbon-attached filaments is widely accepted in the process of SV transport and replenishment [64,115,116,118,122,125,128,130,136]. In a recent study, the effect of different activity-states on the number and location SVs as well as their association with filaments is investigated [122]. Here, IHC ribbon synapses are either stimulated with 50 mM K^+^, inhibited or incubated in a resting buffer with subsequent HPF/FS and ET. Notably, at resting ribbon synapses, only one-quarter of all SVs within the first layer around the ribbon is directly attached to the ribbon and stimulation caused a decrease in SV attachment to ribbon. Instead, on average 19% of all ribbon-associated SVs are found to be interconnected to other SVs, and stimulation even triggers an increase of SVs with interconnectors [122]. Additionally, a small proportion of SVs harbors both, interconnectors and ribbon-attached filaments. Stimulation strikingly promotes the presence of such SVs, specifically at the apical ribbon part distant from the AZ membrane [122]. In such a scenario, interconnectors could together with ribbon-attached filaments form a network and this way stabilize the SV around the synaptic ribbon to support their transport towards the AZ membrane.

In summary, interconnectors are present across species, in native [133,134] and near-to-native sample preparations [120,122,132,135,136]. Therefore, they appear to be a key feature of presynapses [138,139,157]. In the future, however, more studies using cryo-ET will be needed to dissect their function in SV recruitment, by using synaptosome preparations [133,134] or neuronal cultures directly grown on EM grids [156,158] from different brain regions and species. Additionally, emerging studies combining cryo-correlative light and EM (cryo-CLEM) [156,158,188], cryo-focused ion beam—scanning EM (FIB-SEM) milling [189,190,191,192] and cryo-ET [157,158] will make it possible to investigate greater 3D volumes of hair cell and retinal ribbon synapses with an enhanced signal-to-noise ratio of unstained native tissue.

## 6. Proteins Potentially Involved in Interconnector Formation of Neuronal Synapses

As pointed out in the previous sections, interconnectors are a common feature of chemical synapses. Thus, the question about their molecular nature arises. In the past, numerous studies on a range of presynaptic protein mutants in different model systems aimed to clarify this issue. Synapsin is one of the most abundant phosphoproteins present on SVs [193,194,195] and is considered as a candidate constituent of interconnectors at neuronal synapses [131,132,134,196,197]. Studies on synapsin knockouts (KOs) [132] have revealed that a meshwork of interconnectors is essential to organize the reserve pool by limiting the diffusion of SVs [131,132,134]. In line with these findings, upon K^+^ stimulation or treatment with the phosphatase inhibitor okadaic acid, the dissociation of synapsin from SVs increases the vesicle mobility and their release in mouse synaptosomal preparations [134,198].

Moreover, Milovanovic et al. [199] offered the hypothesis that synapsin establishes and self-segregates into a distinct liquid phase with or without the assistance of other scaffolding proteins at a synapse. This separated synapsin1 can capture SVs through its C-terminal intrinsically disordered regions, which allow SVs to form clusters, but at the same time ensures SV movement within the presynaptic terminal. Noteworthy, a synapsin independent subset of interconnectors seems to exist as well, because in synapsin triple KO mice only 60% of the interconnectors could be abolished as observed in synapses of hippocampal slice cultures [132]. Moreover, F-actin has been speculated to act as a scaffold for SV tethering [132], and this way regulates SV mobility [200]. However, experiments involving depolymerization of F-actin did not confidently approve this hypothesis [201,202,203,204]. Therefore, the components of interconnectors at neuronal synapses are still largely unknown.

## 7. Proteins Potentially Involved in Interconnector and Ribbon-Attached Filament Formation at Ribbon Synapses

The synaptic ribbon is a highly specialized structure with a unique protein composition. Thus, it remains questionable, whether the filaments found at ribbon-type synapses are homologous or analogous to those present at neuronal synapses. Synapsins that have been discussed in the context of interconnectors for neuronal synapses are absent in ribbon synapses [205,206]. Moreover, F-actin is proposed to regulate exocytosis at murine IHC ribbon synapses [207,208], by limiting the access of SVs to the release sites [208]. However, direct evidence for actin serving as a filament component is still lacking for ribbon synapses.

A few putative candidate proteins are discussed in the literature that could potentially be involved in filament formation at the ribbon. The major scaffolding component of the synaptic ribbon is the vertebrate- and ribbon-specific protein RIBEYE [69]. RIBEYE consists of two domains, the ribbon-specific A domain maintains the structural organization of the ribbon and the B domain is homologous to the C-terminal binding protein (CtBP2), which functions as a transcription repressor in most tissues [69]. The B domain exhibits enzymatic activity [209] and is thought to participate in SV attachment to the ribbon [69,210,211]. Due to the complete loss of synaptic ribbons in RIBEYE KO mice at retinal photoreceptors [212] and IHCs [120,213], it is challenging to single out RIBEYE as a direct component of ribbon-attached filaments. In zebrafish neuromast hair cells, after ribeye a/b deletion still a halo of SVs is present around ribbon-sized structures lacking the typical electron-density. These ‘ghost-ribbons’ are somewhat smaller in size but are surrounded by a comparable number of SVs as in control animals. Moreover, ghost ribbons show residual ribeye signal with immunofluorescence [214], suggesting a reminiscent amount of ribeye could potentially still link SVs.

As stated earlier, the ribbon can efficiently mobilize SVs, which is thought to involve ribbon-attached filaments [116,125] as well as interconnectors [122]. In this regard, the ATP-dependent, active vesicle transport theory is mainly prompted by the identification of a kinesin polypeptide (Kif3a) at the murine photoreceptor ribbon synapses by post-embedding immunogold EM [85,215], while microtubules (MT), the cytoskeleton pathway utilized by kinesin motors, directly at membrane-anchored ribbons are missing at photoreceptor ribbon synapses [215]. Interestingly, freeze-etching/FS of frog photoreceptor ribbon synapses revealed proteinaceous knobs (30–50 nm) that decorates the ribbon surfaces. These have been suggested to function as ‘stepping stones’ on which motor proteins might stilt walk [130].

In hair cells, MTs terminate close to membrane-anchored ribbons [117,124]. Moreover, MTs have been detected associated with purified ribbons [216]. MTs can further be observed nearby cytoplasmically floating ribbon precursors in pre-hearing mice [124]. Additionally, super-resolution stimulated emission depletion (STED) microscopy revealed a close association of Kif1a, but not Kif2a and Kif5 with these floating ribbon precursors in immature IHCs [124]. Therefore, kinesin dependent MT transport of SV remains an option at ribbon-type synapses, though the polarity of the MT in IHCs is not known yet.

Another candidate protein that could mediate potential active SV transport along the ribbon to the AZ membrane is the actin-based motor protein myosin. Membrane capacitance measurements from Myo6 KOs revealed reduced exocytosis at murine IHC ribbon synapses [217]. Further, post-embedding immunogold EM confirms the presence of Myo6 directly at the ribbon [217] along with an AZ membrane-specific labeling, as also shown for the Myo6 interacting protein otoferlin [97,98,217]. Yet, SVs remain attached to the synaptic ribbon in both, otoferlin as well as Myo6 KOs [90,97,217], thereby ruling these proteins out as the main component of interconnectors or ribbon-attached filaments. Nonetheless, the potential role of otoferlin in SV tethering to the AZ membrane [90] will be discussed in the next sections.

Finally, at hair cell and photoreceptor ribbons, only the short isoform of piccolo, called piccolino, is expressed, which lacks the C-terminal part of full-length piccolo [218,219]. Consequently, binding sites for bassoon and rab3-interacting molecule (RIM) are absent [219]. Immunogold labelings against bassoon and the N-terminal of piccolo show clear spatial segregation of these proteins at ribbon synapses, whereby piccolino, unlike bassoon, is found directly at the ribbon as described for retinal and IHC ribbon synapses in mice [16,37,87,124,127]. The reduction or absence of piccolino at ribbon synapses drastically influences the ribbon size and shape in the retina [127,218]. Further, it has been shown that piccolino interacts with RIBEYE molecules via multiple PxDLS-like motifs within the piccolino protein, arranging both molecules in a highly structured way [127]. In the proposed model, the piccolino N terminus is facing the cytoplasm, making the organization of other proteins, that are, for example, involved in SV cycling, possible [127]. Thus, speculatively, RIBEYE-piccolino interaction could assist the attachment of SVs to the ribbon. Due to bassoon’s localization closer to the AZ membrane at ribbon-type synapses [16,85], the role of bassoon in SV tethering will be discussed in the following sections that deal with membrane-proximal SVs.

Thus far, we pointed out that advanced EM methods provide the required resolution to decipher the finer structural details at synapses. The prevailing view in the field proposes that two major categories of filaments at SVs exist, which defines SV function at synapses. At both synapses, neuronal and ribbon synapses, SVs away from the AZ membrane are highly interconnected and form a network of SVs. We propose that interconnectors and in the case of ribbon synapses, additionally ribbon-attached filaments, organize SV pools and determine SV availability.

## 8. SV Tethering in Membrane-Proximity

Next to the presence of interconnectors, also tethering of SVs at the membrane appears to be a unifying feature between neuronal and ribbon-type synapses [16,90,120,121,122,125,132,133,134,135] in order to recruit SVs to the membrane and mediate release [122,125,133,134,135,150,151]. We previously discussed that the SVs near the AZ membrane are frequently defined as the RRP of SVs at the ribbon-type synapses [12,99,116]. At neuronal synapses, specifically docked SVs are shown to account for the RRP [150,151], which is only a small proportion of all SVs near the membrane. Other SVs are either linked to the AZ membrane via single or multiple tethers. In the coming sections, we will first discuss membrane tethering and subsequent docking for neuronal synapses together with potential proteins involved in this process. In a later section, we will provide a comparison to ribbon-type synapses.

## 9. Synaptic Vesicle Tethering at the AZ Membrane of Neuronal Synapses to Recruit SVs

Different EM methods so far have verified the presence of single—but also multiple-membrane-tethered SVs at vertebrate neuronal synapses [132,133,134,135,137,150], with a broad tether length ranging between 5–120 nm (Figure 2C,D) [129,132,133,134,135,150].

RIM1 supposedly plays a prominent role in membrane tethering of SVs at neuronal synapses. Cryo-ET on murine vitrified synaptosomes showed that the number of tethered SVs is significantly reducing in RIM1α KOs, where RIM1α might specifically contribute to the formation of tethers of ~10 nm length [133]. Moreover, molecular analysis has shown that in RIM1α KOs the priming factor Munc13-1 is down-regulated [133,220,221,222]. Munc13-1 assists explicitly in recruitment and priming of SVs downstream of RIM at neuronal synapses [223]. Interestingly, the number of SVs 45 nm away from AZ is reduced in RIM1α KO [133]. Conclusively, RIM1α and/or Munc13-1 are crucial players in tether formation in neurons [133,224], but likely only RIM might also be involved in the initial tethering step, employing long, single tethers [133], to bring SV near to the AZ membrane.

Moreover, the two large scaffolding proteins of the cytomatrix at the AZ, piccolo, and bassoon [225,226,227,228] could mediate SV tethering. They seem to fulfill a redundant role at neuronal synapses, and both are found to organize SVs at AZs [226]. Investigations on the endbulb of Held on either bassoon [229] or piccolo [230] KOs revealed that the individual protein disruption had slightly different functional consequences for these synapses, but in both studies, a significant reduction of SVs in the vicinity of the membrane has been described on the ultrastructural level [229,230]. This qualifies both proteins as potential tether constituents, but only the morphological analysis of tethers within the individual mutants using HPF/FS together with ET would enable to draw a conclusion on their contribution to tether formation.

Finally, tethers are described to ensure a tight coupling distance of SVs to Ca^2+^ channels, as proposed for the chicken ciliary calyx [231,232,233]. In chicken synaptosomal preparations it has been shown that a Ca_V_2.2 C-terminal-dependent SV tethering takes place at the AZ to recruit SVs from the nearby cytoplasm and to place them close to the Ca^2+^ channel to permit single domain gating [234,235]. Future studies will be required to determine, whether this could also account for other types of Ca^2+^ channels.

## 10. SV Docking and Priming at Neuronal Synapses

As a final preparatory step before fusion, morphological docking takes place at presynaptic AZs [150,151]. At neuronal synapses, SV docking is possible by the force-generating interactions between the SV membrane-protein synaptobrevin and the plasma membrane proteins syntaxin and SNAP 25. These interactions allow for the formation of the SNARE core complex and are regulated by auxiliary proteins [18,236,237,238,239,240]. Moreover, the vesicular protein synaptotagmin-1 serves as the primary Ca^2+^ sensor [11,241,242,243,244,245,246,247,248,249,250] to regulate intimately the SNARE zipping of SVs to the AZ membrane [2,167,238,240,251,252,253,254]. Some recent advances in single particle Cryo-EM and X-ray crystallography provide a near-atomic resolution of SNARE protein complex structure [236,255,256,257,258]. At this moment, the crystal structure of a tripartite interface between synaptotagmin-1, the SNARE complex, and the protein complexin is available [259]. This tripartite complex forms at docking sites for SVs and has to become unlocked to trigger fusion [259]. Docked SVs are primed to the AZ membrane upon arrival of an action potential to undergo synchronous fusion due to the increase in the intracellular Ca^2+^ concentration. The interaction of priming factors of the Munc13 protein family and the SNARE complex is essential for this step [260,261,262,263]. Munc13 and Munc18-1 together mediate the SNARE complex assembly [263,264,265,266], and RIM appears to assist in the process of priming [267]. Furthermore, RIM family proteins not just interact with Munc proteins, but also with Ca^2+^ channels and rab3 [220,268,269,270,271] to define the RRP size [221,223,268,269,270,271,272,273].

Based on studies of a plethora of synaptic models the consolidated view has emerged that at resting synapses only a small population of SVs are in direct contact to the AZ membrane. The fraction of docked SVs is frequently proposed to represent the RRP [39,150,151,169]. Physiologically, SVs in the status of priming are defined as the RRP, because of their ability to instantly fuse with the plasma membrane in response to an occurring stimulus [151,167,183]. Moreover, in hippocampal neurons, Liprinα3 colocalizes with other AZ proteins like bassoon, RIM, Munc13, RIM-binding protein, and ELKS [274]. In Liprinα3 KOs, a loss of docked SVs has been observed after CAF as well as HPF/FS, but also a reduction of 25% of all SVs within 100 nm from the AZ membrane is reported. Therefore, the authors concluded that Liprinα3 also plays a role in SV tethering [274].

In the past, the analysis of random ultrathin sections limited the identification of membrane-to-membrane contacts of a SV to the AZ, the determination of spaces within a nanometer range is not possible in ultrathin sections of 50-100 nm due to the poor z-resolution [151]. Therefore, classical analyses employed criteria where SVs within a certain distance of the AZ, i.e., 30 [275], 40 [276], and 50 nm [277], have been defined as docked, assuming that SV populations within these distances are functionally homogeneous [151].

Consequently, the definition of a morphologically docked SV is not consistent throughout literature. Taking into consideration that CAF based preparation methods might induce shrinking artifacts, Schikorski and Stevens [278] carefully coined the term ‘AZ-proximal SVs’. Using this approach, ~10 AZ-proximal SVs are estimated at CA1 excitatory synapses of the mouse hippocampus. The use of ET enables a higher spatial resolution making a clear definition of membrane attachment possible [151,154]. Relating structure to function by inducing precise and short optogenetic stimulations instantly followed by HPF (Opto-HPF) in cultured hippocampal neurons reduces the number of morphologically docked SVs at the AZs, which therefore indeed represent the RRP at these synapses [152].

Furthermore, freezing methods circumvent the immobilization via fixatives, which avoids artifacts of protein agglomeration and subsequently shrinkage due to dehydration. This way, HPF/FS and ET reveal docked SVs in tight AZ membrane contact [132,150,151,274] (Figure 2F,F’). In 200 nm thick sections of wild-type mice, ~10–12 docked SVs per synapse, at a distance of 0–2 nm to the membrane, are found in organotypic cultures after HPF/FS and ET [151]. In thinner, 100 nm sections of spine synapses, ~3–4 docked SVs have been counted [150]. Additionally, studies on synapses of hippocampal slice culture have shown small tethers linking docked SVs to the presynaptic membrane. In line with these findings, multiple-filaments or ‘strands’ (Figure 2E,E’) have been observed in the vicinity to docked SVs derived from hippocampal neurons in acute slice preparations as well as dissociated neuronal cultures [132,135]. Similar multiple-tethers have been observed in the native, still hydrated cortical synaptosomes with cryo-ET. Here, stimulation depletes a subset of membrane-proximal SVs with 2.5 nm long tethers [133,134], which prompted the definition of docking in a range of 0–2 nm distance [151]. Indeed, X-ray structural investigations of the SNARE complex revealed that helical extensions reach into the cytoplasm [279], which supports the assumption that these short tethers are formed by the SNARE proteins.

To further dissect the role of presynaptic proteins in priming and docking, mutants for important key players have been investigated using HPF/FS and ET in several laboratories. The deletion of the priming factors of the Munc13 protein family results in the loss of docked SVs [53,150,151]. A similar defect is observed for CAPS1/2 double-KOs as well [151], fitting with the complete absence of release as observed in electrophysiological recordings of mutants lacking the Munc13-1 and Munc13-2 [280] or CAPS double-KOs [281]. Furthermore, Imig et al. [151] have investigated the number of docked SVs in single-KOs for different SNARE-complex proteins, priming factors, synaptotagmin-1, and complexins. In this study, a striking reduction of docked SV numbers has been observed in Munc13/CAPS mutants. SNARE protein KO synapses also exhibit a severe reduction in docking, while the absence of complexins and synaptotagmin-1 had no influence on the number of docked SVs. These findings are in agreement with the RRP size estimates obtained from physiological measurements [282,283,284]. These results finally led to the interesting conclusion that morphological docking, functional priming and SNARE complex assembly represent the same process [151].

In summary, at neuronal synapses, initial SV tethering to the AZ might involve proteins such as RIM1α, but also piccolo, bassoon, and Liprinα3 at neuronal synapses, while shorter tethers appear to consist of SNARE complex proteins and/or Munc13/CAPS family proteins.

## 11. Synaptic Vesicle Tethering at the AZ Membrane of Ribbon Synapses Recruit SVs

At ribbon-type synapses, membrane-attached tethers have been first reported using CAF at retinal and IHC ribbon synapses [16,125] and then by using HPF/FS, especially at IHC ribbon synapses [90,120,121,122] (Figure 2I,J). In IHCs, most of the tethered SVs are located laterally close to the presynaptic density of the ribbon, while their occurrence reduces with increasing lateral distance [16,90,122]. Unlike for neuronal synapses [132,135,137], the size distribution of tethers is less broad at IHC ribbon synapses and ranges between 10–50 nm [16,90,122] with an average of ~23 nm at rest [122]. However, a few short tethers below 10 nm are encountered at IHC ribbon synapses as well [90,122]. Due to this observed diversity in tether appearance and length across different preparations, it is likely that the interplay of several different proteins induces tether formation. Moreover, the involved key molecules may differ depending on the ribbon-synapse type. Yet, the identity of these tethers is entirely unknown.

In recent ET studies, membrane-proximal SVs are defined in a range 100 nm laterally from the presynaptic density and extending up to 50 nm in z, due to the observed tether length of up to ~50 nm at the membrane [90,120,121,122]. In these studies, a detailed characterization has been performed for the localization and position of SVs with and without tethers in wild-type during different activity states. Distinct SV localizations and association with tethers have been observed at IHC ribbon synapses [90,120,121,122]. Moreover, striking shifts in the proportion of tethered and non-tethered SVs have been reported [122], while a surprisingly stable SV number in membrane-proximity under rest, stimulation or inhibition led to the hypothesis that distinct SV sub-pools do exist within the morphological membrane-proximal pool at these synapses [122].

Moreover, at murine IHC ribbon synapses, SVs are also attached to the presynaptic density via single tethers [90,120,121,122] of ~20 nm length [122] (Figure 2H). Prolonged stimulation reduces the fraction of these SVs, while more SVs with single tethers, attached directly to the AZ membrane, are present. Suggestively, SVs move from the ribbon to the membrane, via an attachment to the presynaptic density [122] (Figure 2G).

## 12. Potential Components of SV Tethers at IHC Ribbon Synapses

While at neuronal synapses priming factors and SNARE proteins likely play a significant role in tether formation, at ribbon-type synapses the situation is still elusive, especially for IHC ribbon synapses. However, at ribbon synapses, bassoon plays an essential role in anchoring the synaptic ribbon in IHCs [16,86,88] and the retina [85,87], while piccolo, appearing only in its short isoform piccolino at ribbon-type synapses [124,127,218,219] is spatially segregated from bassoon and localizes to the ribbon itself [37,124]. An intact synaptic ribbon and its proper anchorage, mediated by bassoon, has been proposed to promote the RRP of SVs via clustering of Ca^2+^ channels at the release sites [16,74,86,87,120,212,213]. At bassoon deficient IHC ribbon synapses the clustering of Ca^2+^ channels has been significantly impaired [16,86,88]. Consequently, the number of docked SVs at the AZ of IHC ribbon synapses is reduced in bassoon mutants [16,86]. Upon deletion of the protein RIBEYE, at murine IHC ribbon synapses even multiple presynaptic, bassoon containing, densities are formed that could tether several SVs [120], in numbers quite similar to the wild-type situation [120,122]. Conceivably, bassoon might take over a prominent structural role for a subset of tethers at IHC ribbon synapses and potentially also other ribbon-types synapses.

In contrast to neuronal synapses, where RIM1α is suggested as a tether component [133], at ribbon-type synapses, only RIM2α is present [121,285], but either directly or indirectly involved in tether formation at IHCs [121]. A study using HPF/FS revealed that in these KOs a significant reduction in the number of tethered SVs takes place, but with a normal tether length (~23 nm) [121]. Importantly, the multi-C_2_ domain protein otoferlin plays a crucial role in hair cell exocytosis [81,90,97,98,99,100,101]. At otoferlin KO IHC ribbon synapses the fraction of SVs with short tethers (<10 nm) has been reduced by two-thirds [90], indicating a function of otoferlin in membrane-proximal tether formation as well, but likely in a different process than RIM2α. However, both KOs could not eliminate all tethers at the AZ membrane of these IHC ribbon synapses, leaving the nature of the tethers still unknown.

## 13. Synaptic Vesicle Docking at Ribbon Synapses

As for neuronal synapses also for ribbon-type synapses the definition of docked SVs differs depending on the embedding method and whether ET or ultrathin sections have been used. As mentioned above, often, all membrane-proximal SVs are thought to represent the RRP [81,86,98,114,116,173,175,176,177]. This notion is also supported by ultrastructural studies, where the number of membrane-proximal SVs matches estimates of cell-physiological analyses [77,81,86,173,286]. In contrast, some structure-function studies have reported depletion of only a fraction of membrane-proximal SVs upon stimulation [115,122,176].

Incorporation of HPF/FS at murine retinal ribbon synapses could show that several docked SVs are in physical contact with the AZ membrane [128], consistent with the definition of morphologically docked SVs in neuronal synaptic preparations [150,151,154]. In retinal ribbon synapses, complexin and Munc family proteins can be found [92,93,94,95,128,287]. They employ complexins 3 and 4 [92,93,94,95] that are structurally and functionally distinct from complexin 1 and 2 [94], which are present at neuronal synapses [288,289,290,291]. Complexins have been recently suggested to be involved in SV tethering at ribbon synapses of rod bipolar cells, which showed a reduced number of SVs near the membrane and the ribbon [292]. Surprisingly, at the Munc13-2 deficient photoreceptor ribbon synapses, docked SV numbers are unperturbed. The physiological phenotype in the Munc13-2 KO also appears to be moderate; thus, cumulatively suggesting that photoreceptor ribbon synapses and neuronal synapses differ in terms of Munc13 dependent SV priming [128].

Noteworthy, at IHC ribbon synapses, docked SVs (Figure 2K,K’) are rather rarely observed [122], most membrane-proximal SVs are found to be tethered [90,120,121,122], and therefore still have a certain distance to the AZ membrane. Docked SVs might be extremely short-lived, explaining for their rare appearance considering the exceptionally rapid vesicle turnover rates at IHC AZs [81,98]. Functional studies demonstrated that the kinetics of fusion of the RRP at ribbon synapses could be extremely rapid within 30 ms, depending on the ribbon-type synapse, experimental conditions and analysis [12,60,81,105,173,207,286,293,294,295,296]. Moreover, paired recordings reported two discernible kinetic components of the RRP at IHCs itself [60,297], supporting the presence of heterogeneous SV populations with different release competencies. As discussed in the previous sections, at IHC ribbon synapses, many classical AZ proteins are lacking. Munc13s and CAPS mutant do not show any effect on SV exocytosis at IHC ribbon synapses [90]. Moreover, exocytosis in hearing mice seems to function without the neuronal SNARE complex formed by syntaxin 1, VAMP 1 and 2, and SNAP 25 at IHC ribbon synapses. The study on these KOs or application of botulinum neurotoxins, targeting SNAP 25, syntaxin 1–3, and VAMP 1–3, revealed unperturbed exocytosis at mice IHCs. Further, neuronal SNAREs could not be observed in IHCs using immunostainings [91], although several studies have reported the presence of SNARE mRNA [91,206,216]. This way many candidate proteins that might mediate close SV tethering leading to docking in the membrane seem not to be available at IHC ribbon synapses. Moreover, SNARE regulators such as synaptotagmins 1–3 [298,299] and complexins [96,216] are absent from mature IHCs. Nevertheless, 1-3 tethers per SVs are encountered close to the membrane at ribbon-type synapses in general and IHC ribbon-synapses specifically [16,90,120,121,122,125] (Figure 2J,J’). Interestingly, in RIM-binding protein (BP) 2 mutant IHC ribbon synapses the distance of membrane-proximal SVs is altered, as shown with CAF and ET [300]. Conclusively, it is intriguing that a partially distinct protein composition at ribbon synapses might play a role in tethering.

To summarize, at ribbon synapses, the molecular identity of the tethering components is grossly unknown and distinct from the neuronal synapses. Some of the putative proteins involved in tethering near AZ membrane consists of RIM2α, bassoon, complexin3/4, and the hair cell-specific protein otoferlin. Strikingly, SV docking at IHC ribbon synapses seems to operate without the neuronal SNAREs, implicating an alternative docking and priming mechanism, which is distinct from that of the neuronal synapses.

## 14. Conclusions

Our understanding of the synaptic molecular architecture regulating neurotransmission has expanded based on many structure-function analyses in diverse vertebrate and invertebrate synapses. SV pools are extensively studied using EM on synaptic preparations of *Drosophila* and *C. elegans* NMJs; frog and zebrafish NMJs; different CNS synapses such as from the hippocampus or the calyx of Held, as well as sensory ribbon synapses. In this review, we have focused mostly on the differences and similarities between central nervous system neuronal and ribbon synapses—as summarized in Figure 3 and Table 1.

Despite significant disjointed contributions, the field-specialists are yet to succeed in developing appropriate synergies towards identifying the sequence of exocytosis at neuronal and ribbon synapses. Nevertheless, it is gradually becoming clear that the filaments within the presynaptic terminal might be regulating different stages of exocytosis of ribbon and neuronal synapses.

A simplified proposal is that the interconnectors at the neuronal synapses cluster SVs distant from the AZ membrane, and synapsins might assist in this process (Figure 3A1). Moreover, the interaction of synapsin with other proteins like actin or piccolo helps in SV mobility towards the AZ membrane (Figure 3A2). Although at ribbon synapses the proteins involved in interconnector formation are mostly unknown, the main ribbon component RIBEYE potentially due to its interaction with piccolino, might mediate SV organization. Interconnectors and ribbon-attached filaments together support the organization of the ribbon-associated SV pool and then move the SVs to the AZ membrane at ribbon synapses, which also potentially involves cytoskeletal players such as myosin (Figure 3A1′). In both types of synapses, tethering of SVs at the AZ membrane is common. A single long tether possibly mediated via RIM1α recruits the SVs close to the neuronal AZ membrane (Figure 3A3). After that, several proteins orchestrate together to ensure close coupling of the SVs to the membrane by forming multiple shorter tethers (Figure 3A4). It is an assumption that single and multiple tethers at the membrane consist of different proteins. However, it remains elusive, whether the protein candidates are involved in one or several morphologically distinct, exocytosis steps, as indicated by listing them multiple times. Finally, the docking of SVs involving neuronal SNARE proteins takes place (Figure 3A5).

Analogous sequence preparing SVs for release is taking place in the ribbon synapses as well. Clearly, ribbon-type synapses and specifically IHCs employ distinct molecular players in the tether formation. Under this situation most likely bassoon probably plays a role for bringing the SVs closer to the AZ membrane (Figure 3A2′), after that, RIM2α and otoferlin might regulate the process of SV tethering at AZ of IHCs in distinct pathways (Figure 3A3′). Given that, the SV replenishment rates are extremely fast, the temporal resolution of many of the EM sample preparations could only capture the multiple-tethered SVs (Figure 3A4′) and docking (Figure 3A5′) sporadically at IHC ribbon synapses. However, at the retinal ribbon synapses, the docking of SVs is common. Thus, divergences in the molecular identity mediating release between ribbon-type synapses and between neuronal synapses do exist (See Table 1 for the putative molecular players involved).

## Figures and Tables

**Figure 1 ijms-20-02147-f001:**
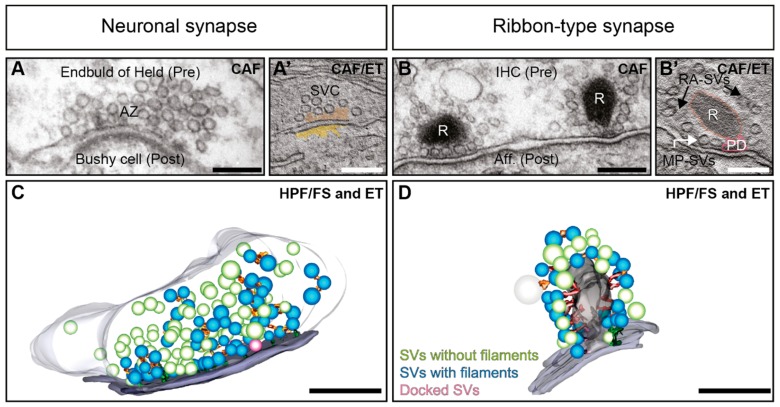
The nanoarchitecture of an excitatory neuronal and ribbon-type synapses. (**A**) Electron micrograph of an individual active zone (AZ) from a mature murine endbulb of Held (presynaptic compartment, Pre) in the anteroventral cochlear nucleus projecting onto a Bushy cell (postsynaptic compartment, Post). (**B**) Electron micrograph of ribbon (R) synapses formed by inner hair cells (IHC, Pre) and afferent fibers (Aff., Post). (**A’**) Example virtual section obtained from electron tomography (ET) on samples prepared by conventional aldehyde fixation (CAF), showing synaptic vesicle clusters (SVC) at the AZ (brown). (**B’**) At ribbon synapses, two morphological SVs pools are present. The ribbon-associated (RA)-SVs (with black arrows) are arranged in a halo around the synaptic ribbon (R, red outline). The membrane-proximal (MP)-SVs (with white arrow) are located near the AZ membrane around the presynaptic density (PD, pink outline). (**C**,**D**) Tomogram models rendered from high-pressure frozen and freeze substituted (HPF/FS) synapses allow the visualization of SVs and tethering in 3D at a near-to-native state. Delicate filaments associated with SVs have been investigated using these methodologies. SVs with filaments (blue) and without filaments (green) are shown, along with morphologically docked SVs (magenta) at a neuronal (**C**) and a ribbon-type synapse (**D**). All scale bars are 200 nm. (**A**,**A’**) and the tomogram for the 3D model in **C** are kindly provided by Anika Hintze, Institute for Auditory Neuroscience, University Medical Center Göttingen.

**Figure 2 ijms-20-02147-f002:**
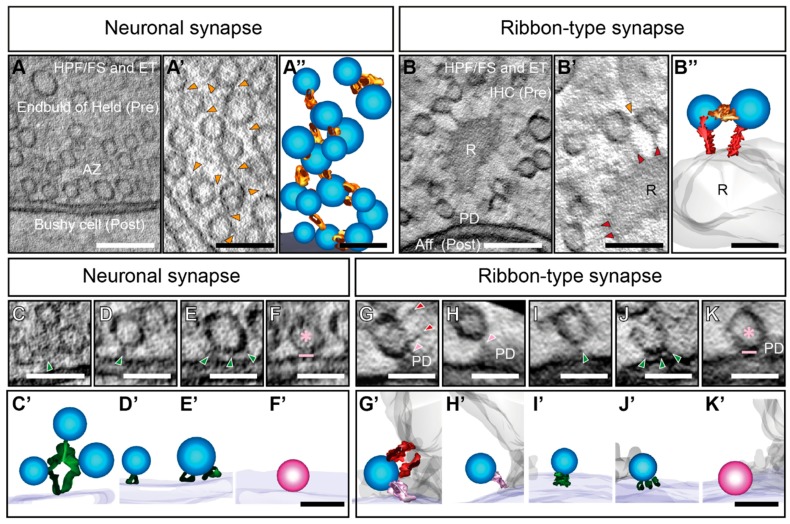
The diversity of filaments and corresponding vesicles at neuronal and ribbon-type synapses. (**A**,**B**) Virtual sections are showing overviews of a murine endbulb of Held active zone (AZ) (**A**) and IHC ribbon (R) synapse (**B**), prepared by HPF/FS. Scale bars 200 nm. Magnifications highlight (**A’**) SV in the cytoplasm and (**B’**) RA-SVs distal from the AZ membrane in the respective synapses. In both cases, filaments called interconnectors (orange arrowheads, (**A’**,**B’**)) connect adjacent SVs. Scale bars 100 nm. In neuronal synapses, these interconnectors (orange) often form chain-like configurations, ensuring clustering of SVs (**A’’**). At ribbon synapses, additional filaments called ribbon-attached filaments (red arrowheads, B’) are present. They mediate the direct association of SVs to the synaptic ribbon (R). In many cases, a single SV can harbor both interconnectors (orange) and ribbon-attached filaments (red) (**B’’**). Scale bars 50 nm. (**C**–**K**) Close-up of SVs at the AZ membrane that are associated with filaments called membrane-attached tethers (green arrowheads). These membrane-attached tethers are of variable lengths and numbers in neuronal (**C**–**E**) and ribbon-type (**I**–**J**) synapses. Additionally, at both synapse types, morphologically docked SVs are present (pink asterisk, (**F**,**K**)). At ribbon synapses, membrane-proximal (MP)-SVs are rarely additionally attached to the synaptic ribbon (**G**). IHC ribbon synapse SVs are frequently attached to the presynaptic density (PD) at the base of the ribbon via presynaptic density-attached tethers (pink arrowheads, H). Scale bars 50 nm. Respective 3D models are shown in (**C’**–**K’**). SVs (blue) with membrane-attached (green), presynaptic density-attached (pink) and ribbon-attached (red) tethers are shown. Docked SVs are depicted in magenta. Scale bars 50 nm. (**A**–**A’’**); (**C**–**F**) and the tomogram for the 3D models in (**C’**–**F’**) are kindly provided by Anika Hintze, Institute for Auditory Neuroscience, University Medical Center Göttingen.

**Figure 3 ijms-20-02147-f003:**
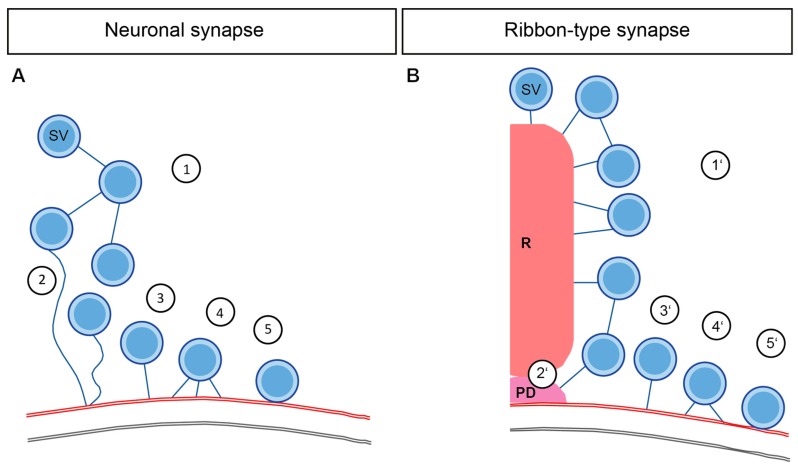
Schematic comparison of potential filament protein candidates at neuronal and ribbon-type synapses. (**A**) At neuronal synapses, the interconnectors cluster SVs away from the AZ membrane (1). Initially, a single long membrane-attached tether mobilizes the SVs (2) and recruits SVs to the AZ membrane (3). Multiple short membrane-attached tethers are then formed (4), preparing SV for docking (5) and fusion. (**B**) At the ribbon-type synapses, interconnectors and ribbon-attached filaments are formed, organizing SVs away from the AZ membrane, and then mobilize SVs (1′); possibly via the presynaptic density as suggested for the IHC ribbon synapse (2′). Single membrane-attached tethers are formed (3′) before multiple-tethering (4′), and finally, SVs are docked (5′), and release takes place.

**Table 1 ijms-20-02147-t001:** Possible candidate proteins directly or indirectly involved in filament formation at different stages of exocytosis. Depicted are candidate proteins, which could assist in the formation of different filaments like interconnectors, ribbon-attached filaments and membrane tethers (also shown in Figure 3). Noteworthy, for most of the studies, it is unknown, whether the protein candidates are an integral part of the filaments or only indirectly influence the filament number and amount of connected SVs.

Neuronal Synapse		Ribbon Synapse	
**Step 1**	Synapsins [131,132,134,135,196]	**Step 1′**	RIBEYE-Piccolino [127]
	F-Actin [132,200,201,202,203,204]		RIBEYE [69,210,211]
			Myosin6 [217]
			F-Actin [200,207,208]
**Step 2**	Bassoon and Piccolo [226]	**Step 2′**	Bassoon [16,120,122]
	F-Actin [132,200,201,202,203,204]		
	Liprinα3 [274]		
**Step 3**	RIM1α [133] Liprinα3 [274]	**Step 3′**	RIM2α [121,285]
			Complexin3 (excluding IHC ribbon synapses) [292]
			RIM-BP2 [300]
**Step 4**	RIM1α [133]	**Step 4′**	Otoferlin (only in HC ribbon synapses) [90]
	Munc13 [133,224]		
	Liprinα3 [274]		
	Ca_v_2.2 channels [234,235]		
**Step 5**	Munc13 [133,150,151,224]	**Step 5′**	Otoferlin (only in HC ribbon synapses) [90,97]
	Liprinα3 [274]		
	Neuronal SNAREs [135,151]		

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
