# Peer review of "Nanomachinery Organizing Release at Neuronal and Ribbon Synapses"

_ijms, 2019, doi:10.3390/ijms20092147_

Round 1

Reviewer 1 Report

The manuscript of Chakrabarti and Wichmann is a very nice review about the organization of the SV release machinery at synapses with a focus on ribbon synapses. Mainly two types of information are presented: a) molecular architecture of structures based on ultrastructural research and b) data from biochemical and functional studies that identified concrete molecular identities, such as piccolo, RIM1a etc. In following paragraphs a synthesis then brings together ultrastructure and molecular identity and integrates this into a model where the arrangement of the molecular components is mapped onto the ultrastructural features. The quality of text, references, figures and layout is pretty good and so I would have only minor suggestions. First, a small comment on formatting: at a couple of instances the font type and size vary. But that can be easily matched in the editorial process. Second, the authors compare the ribbon synapses, which appear to be their prime focus, to CNS synapses. Maybe, for the matter of comparison and to understand consistencies, it would be useful, to incorporate in Figure 3 and Table 1 also the neuromuscular junction, where McMahan and colleagues have also proposed many of these ultrastructural correlates. But that is just a suggestion.

Author Response

Point-to-point response:

Reviewer 1

Comments and Suggestions for Authors

The manuscript of Chakrabarti and Wichmann is a very nice review about the organization of the SV release machinery at synapses with a focus on ribbon synapses. Mainly two types of information are presented: a) molecular architecture of structures based on ultrastructural research and b) data from biochemical and functional studies that identified concrete molecular identities, such as piccolo, RIM1a etc. In following paragraphs a synthesis then brings together ultrastructure and molecular identity and integrates this into a model where the arrangement of the molecular components is mapped onto the ultrastructural features. The quality of text, references, figures and layout is pretty good and so I would have only minor suggestions.

We thank the reviewer for the positive comments.

First, a small comment on formatting: at a couple of instances, the font type and size vary. But that can be easily matched in the editorial process.

We thank the reviewer for spotting. The manuscript is now in a different format.

Second, the authors compare the ribbon synapses, which appear to be their prime focus, to CNS synapses. Maybe, for the matter of comparison and to understand consistencies, it would be useful, to incorporate in Figure 3 and Table 1 also the neuromuscular junction, where McMahan and colleagues have also proposed many of these ultrastructural correlates. But that is just a suggestion.

Thank you for the suggestion.

Our initial submission already cited some of the work from McMahan and colleagues. In appreciation of their valuable work, we now have extended the discussion of their findings in the light of other synapses in the text and the conclusion part. We hope that this will prompt the reader to appreciate their beautiful studies. However, we decided to not include their work into Figure 3 and Table 1, as the focus of the review is on excitatory, mainly central synapses and ribbon-type synapses.

Reviewer 2 Report

Review of the manuscript “Nanomachinery organizing release at neuronal and ribbon synapses” by

Chakrabarti  and Wichmann submitted to IJMS Journal.

In spite of enormous attempt to decipher mechanisms of neurotransmission and its regulation many aspects of these processes remain elusive. Our comprehension of synapses is far from being understood, and our understanding of their function depends on results of electrophysiology, biochemistry and molecular biology. Important breakthrough in our knowledge can be achieved due to the fast development of new technologies, including electron tomography combined with a rapid

 freezing immobilization of neuronal samples.  The authors consider how these methods can improve our knowledge about synaptic organization and functions of neuronal and ribbon synapsis. The topic is important and will be interesting for the readers of the IJMS Journal.

The following corrections should be made.

Abstract:

Lines 28-29:“However, both synapse types share the presence of filamentous structures…”

The authors should correct the sentence as follows: “However, both synapse types share the filamentous structures…”

Keywords: The authors should add “electron tomography” to the list of keywords.

Introduction:

Lines 39-40.

“Neurotransmission relies on Ca2+-dependent exocytosis of SVs, which takes place at dedicated AZs within the presynaptic terminals of neurons [1–4]…”

The authors should add the following reference after [1–4]: Surgucheva et al., “g-Synuclein: Seeding of α-Synuclein Aggregation and Transmission Between Cells”. Biochemistry, 2012; 51:4743-54

Line 47.

“Similar to vertebrate synapses, also invertebrate synapses often show electron-dense projections…”

Should be corrected as follows: “Similar to vertebrate synapses, invertebrate synapses also often show electron-dense projections…”

Line 272. “…reserve pool by limiting the diffusion of SVs [114,115,118]. In line with these findings, upon KCl…”

“by limiting” looks like it is in different fonts. The same is on line 421:” protein complexin are available [245]”. The same is on lines 571-573.

Lines 295. “A few concrete protein candidate proteins are discussed”

This is a clumsy sentence which should be corrected

Lines 301-302 “... However, it is challenging to isolate if RIBEYE as a component of ribbon-attached filaments.

The sense of this sentence is not clear. What the authors mean by “isolate”? Should be corrected.

Figures. Line 112, Figure 1, It is hard to see the apostrophe on A’, it should be presented more clearly.

Legend to Figure 3. The authors should explain what means PD and R

The same concerns the apostrophe on B’.

Figure legends. The authors should explain to what the arrows on B and B’ are directed.  

Figure 1. Line 117-119. “(B’) At ribbon synapses, two morphological SVs pools are present. The ribbon-associated (RA)-SVs are arranged in a halo around the synaptic ribbon (R, red). The membrane-proximal (MP)-SVs are located near the AZ membrane around the presynaptic density (PD, pink).”

 This figure legend is difficult to understand. The authors explain B’ fragment of the figure and refer to R, whereas Rs are present not on B’, but on B. Further they refer to PD, pink, but again PD is shown on B, but not on B’. Should be corrected.

Author Response

Point-to-point response:

Reviewer 2

Comments and Suggestions for Authors

Review of the manuscript “Nanomachinery organizing release at neuronal and ribbon synapses” by

Chakrabarti and Wichmann submitted to IJMS Journal.

In spite of enormous attempt to decipher mechanisms of neurotransmission and its regulation many aspects of these processes remain elusive. Our comprehension of synapses is far from being understood, and our understanding of their function depends on results of electrophysiology, biochemistry and molecular biology. Important breakthrough in our knowledge can be achieved due to the fast development of new technologies, including electron tomography combined with a rapid freezing immobilization of neuronal samples.  The authors consider how these methods can improve our knowledge about synaptic organization and functions of neuronal and ribbon synapsis. The topic is important and will be interesting for the readers of the IJMS Journal.

The following corrections should be made.

We thank the reviewer for the detailed and constructive comments. We have included the suggested changes in the manuscript and modified the text, figure, and figure legends accordingly.

Abstract:

Lines 28-29:“However, both synapse types share the presence of filamentous structures…”

Done

The authors should correct the sentence as follows: “However, both synapse types share the filamentous structures…”

Done

Keywords: The authors should add “electron tomography” to the list of keywords.

Done

Introduction:

Lines 39-40.“Neurotransmission relies on Ca2+-dependent exocytosis of SVs, which takes place at dedicated AZs within the presynaptic terminals of neurons [1–4]…”

The authors should add the following reference after [1–4]: Surgucheva et al., “g-Synuclein: Seeding of α-Synuclein Aggregation and Transmission Between Cells”. Biochemistry, 2012; 51:4743-54

Thank you for the suggestion.

Through our previous statement, we specifically wanted to highlight the importance of calcium-dependent SV exocytosis at the active zone of neurons. The focus of Surgucheva et al., 2012 is to show how γ synuclein oxidation promotes α synuclein aggregation, and this way is leading to the neurodegeneration. Nonetheless, we fully understand the importance of secretion of synucleins and many other endogenous proteins via exosomes. Therefore, we have now rephrased our introductory lines. Some additional citations are included in addition to Surgucheva et al. to support these statements.

Line 47.“Similar to vertebrate synapses, also invertebrate synapses often show electron-dense projections…”

Should be corrected as follows: “Similar to vertebrate synapses, invertebrate synapses also often show electron-dense projections…”

Done 

Line 272. “…reserve pool by limiting the diffusion of SVs [114,115,118]. In line with these findings, upon KCl…”

“by limiting” looks like it is in different fonts. The same is on line 421:” protein complexin are available [245]”. The same is on lines 571-573.

Done 

Lines 295. “A few concrete protein candidate proteins are discussed”

This is a clumsy sentence which should be corrected

Rephrased

Lines 301-302 “... However, it is challenging to isolate if RIBEYE as a component of ribbon-attached filaments.

The sense of this sentence is not clear. What the authors mean by “isolate”? Should be corrected.

Rephrased

Figures.

Line 112, Figure 1, It is hard to see the apostrophe on A’, it should be presented more clearly.

Legend to Figure 3. The authors should explain what means PD and R

The same concerns the apostrophe on B’.

Figure legends.

The authors should explain to what the arrows on B and B’ are directed.  

Figure 1. Line 117-119. “(B’) At ribbon synapses, two morphological SVs pools are present. The ribbon-associated (RA)-SVs are arranged in a halo around the synaptic ribbon (R, red). The membrane-proximal (MP)-SVs are located near the AZ membrane around the presynaptic density (PD, pink).”

 This figure legend is difficult to understand. The authors explain B’ fragment of the figure and refer to R, whereas Rs are present not on B’, but on B. Further they refer to PD, pink, but again PD is shown on B, but not on B’. Should be corrected.

Thank you for the suggestions. We have now modified the figure labelings to fit with the figure legends and also explained the abbreviations in the legend clearly.